# GENERALIST: A latent space based generative model for protein sequence families

**Hoda Akl**[1]*, **Brooke Emison**[2], **Xiaochuan Zhao**[1], **Arup Mondal**[3], **Alberto Perez**[3], **Purushottam D. Dixit**[2,4]*

**1** Department of Physics, University of Florida, Gainesville, Florida, United States of America, **2** Department of Biomedical Engineering, Yale University, New Haven, Connecticut, United States of America, **3** Department of Chemistry, University of Florida, Gainesville, Florida, United States of America, **4** Systems Biology Institute, Yale University, West Haven, Connecticut, United States of America

* Purushottam.dixit@yale.edu (PDD); hodaakl@ufl.edu (HA)

**Data Availability Statement:** The data and the code related to this work is available at https://github.com/hodaakl/GENERALIST/.

**Funding:** This work was supported by NIH grant R35GM142547. PDD received salary support from

## Abstract

Generative models of protein sequence families are an important tool in the repertoire of protein scientists and engineers alike. However, state-of-the-art generative approaches face inference, accuracy, and overfitting- related obstacles when modeling moderately sized to large proteins and/or protein families with low sequence coverage. Here, we present a simple to learn, tunable, and accurate generative model, GENERALIST: *GENERAtive nonLInear tenSor-factorizaTion* for protein sequences. GENERALIST accurately captures several high order summary statistics of amino acid covariation. GENERALIST also predicts conservative local optimal sequences which are likely to fold in stable 3D structure. Importantly, unlike current methods, the density of sequences in GENERALIST-modeled sequence ensembles closely resembles the corresponding natural ensembles. Finally, GENERALIST embeds protein sequences in an informative latent space. GENERALIST will be an important tool to study protein sequence variability.

## Author summary

Protein sequence families show tremendous sequence variation. Yet, it is thought that a large portion of the functional sequence space remains unexplored. Generative models are machine learning methods that allow us to learn what makes proteins functional using sequences of naturally occurring proteins. Here, we present a new type of generative model GENERALIST: *GENERAtive nonLInear tenSor-factorizaTion* for protein sequences that is accurate, easy to implement, and works with very small datasets. We believe that GENERALIST will be an important tool in the repertoire of protein scientists and engineers alike.

## Introduction

Advances in *omics* technologies allow us to investigate sequences of evolutionarily related proteins from several different organisms. Surprisingly, even when the function and structure are conserved, sequences within protein families can vary substantially [1]. This variability is

NIH grant R35GM142547. The funders had no role in study design, data collection and analysis, decision to publish, or preparation of the manuscript.

**Competing interests:** The authors have declared that no competing interests exist.

governed by a combination of factors, including protein stability [2], interaction partners [3], and function [4]. Therefore, it is not feasible to rationalize observed variation in protein sequences using bottom-up mechanistic models.

To understand the forces that constrain protein sequence variability and to identify new protein sequences that perform desired functions, we need methods to probabilistically sample sequences that are likely to result in functional proteins [5]. Generative models of protein families that use multiple sequence alignments (MSAs) are one such approach. These models attempt to learn patterns of covariation between amino acids across different positions and model a distribution over the sequence space that captures aspects of the observed covariation. The Potts model is one of the most popular generative models of protein families [6–8]. Potts model is a maximum entropy model constrained to reproduce positional amino acid frequencies and position-position pair correlations. Even though only 1- and 2-site frequencies are constrained, the model can reproduce higher order covariation statistics. The model is easy to interpret, as it assigns an energy to sequences. In addition to modeling covariance between amino acid positions, Potts models have also been used to rationalize effects of mutations on fitness [9–12], and to predict physical contacts between residues [5].

Despite their tremendous success, there are some issues with the Potts model. For example, the associated numerical inference using gradient descent [13] is computationally intensive as it requires Markov Chain Monte Carlo (MCMC) simulations, potentially limiting their application to small proteins and protein domains ($L\sim100$ residues), unless approximate inference methods are employed [11,14,15]. Moreover, there is no realistic way to tune the model beyond one- and two- position moments, for example, by incorporating multi-position correlations. Also, as we will show below, the Potts model does not reproduce statistics related to the probability density of sequence distances and result in highly unnatural optimal sequences.

Field theoretic approaches [16] can systematically generalize the Potts model by incorporating higher order interaction terms. However, these models can only be trained on *very* small sequences. We note that a recent generalization that combines elements of autoregressive modeling and the Potts model; the autoregressive DCA model (ArDCA) [17], addresses the inference difficulties associated with the Potts model while generating novel sequences that accurately reproduce natural sequence statistics. Finally, the Potts models do not obtain a reduced dimensional representation of data points, for example, akin to a latent space embedding that may be used to identify similarity/differences between sequences.

Deep generative models (reviewed in [18]) are a alternative to Potts models that can potentially incorporate very high order interactions. However, these models require large amounts of training data and lack interpretability. Moreover, while sequencing advances have led to large MSAs, especially for bacterial protein families, many human proteins only exist in mammals and other higher order organisms where the MSA sizes are currently limited by the number of sequenced genomes and ultimately by the total number of mammalian species [19]. Neural network architectures are notorious for being over parametrized, including several hyperparameters for training the networks. In addition, as we show below, variational autoencoders [20], a specific type of neural network architecture, perform worse as compared to other tested models in reproducing the correlations in the data, as well as the Hamming distance distributions for the tested protein families. Finally, deep generative models may not necessarily improve in accuracy with the increasing complexity of the architecture.

As an alternative to these approaches, we present here GENERALIST: ***GENERA**tive non-**LI**near tenSor-factorizaTion*-based model for protein sequences and other categorical data. GENERALIST is an easily implementable, interpretable, and accurate description of protein sequences. In GENERALIST, we obtain a latent space representation for each sequence in the natural multiple sequence alignment. Using these latent variables as temperature-like

quantities, we describe the probabilities of natural sequences using the Gibbs Boltzmann distribution [21,22]. The energies of the Gibbs-Boltzmann distribution are shared across all sequences. The modeler only specifies complexity of the model, i.e. the dimension of the latent space. Both the energies and the latent variables are inferred directly from the data. The latent space dimension is tuned to achieve a user-desired tradeoff between the novelty of generated sequences and the accuracy of the ensemble in reproducing properties of the natural MSAs.

We use GENERALIST to model sequence variability in proteins that span multiple kingdoms of life, alignment sizes, and sequence lengths. We compare the performance of GENERALIST with three other generative models, the Potts model (referred to as adabmDCA [13]), the autoregressive DCA model (referred to as ArDCA [17]), and a variational autoencoder-based model (referred to as VAE [20]). We show that compared to these other models, GENERALIST captures higher order statistics of amino acid covariation across sequences. GENERALIST also predicts conservative local optima that are likely to fold in stable three-dimensional structures. Importantly, the ensemble of sequences generated using GENERALIST most accurately captures the probability density of several inter-sequence distances observed in the natural ensemble. Finally, the latent space representation of sequences naturally identifies sequence subsets with differing higher order covariance statistics. We believe that GENERALIST will be an important tool to model protein sequences and other categorical data.

## Methods

### The Mathematical formalism of GENERALIST

In GENERALIST (Fig 1), we take inspiration from restricted Boltzmann machines (RBM) [23] (reviewed in Mehta et al. [24]) wherein observable "spins" are coupled to hidden variables with their own distribution. We start with a one-hot encoded representation of a multiple sequence alignment of $N$ sequences of length $L$; $\sigma_{anl} = 1$ if the amino acid at position $l$ in the protein sequence indexed $n$ has the identity $a$.

$$\pi_{anl} = \frac{1}{\Omega_{nl}} \exp\left( -\sum_{k=1}^{K} z_{nk} \theta_{akl} \right). \tag{1}$$

In Eq (1), we write the probability of observing amino acid $a$ in position $l$ given a sequence specific latent vector $z_n$, $p(\sigma_{al}|z_n)$ as $\pi_{anl}$. $z_{nk}$ are sequence-specific inverse temperature-like quantities (latent space embeddings), $\theta_{akl}$ are position and amino acid dependent variables, and $\Omega_{nl} = \sum_a \exp(-\sum_{k=1}^{K} z_{nk} \theta_{akl})$ is the partition function that normalizes the probabilities. $\Omega_{nl}$ can be explicitly calculated as a sum over the possible categories in a single position (20 amino acids + gap) without requiring MCMC simulations, which is a salient feature of GENERALIST. We note that in GENERALIST, each sequence in the training data gets assigned a latent space embedding. And that the probabilities of individual amino acids in different positions are dependent on this latent space location. This way, GENERALIST automatically incorporates genetic background in evaluating probabilities of mutations. At the same time, when the latent variables are fixed, the probabilities at different positions are independent of each other (while dependent on the latent space location). This means that given a fixed location in the latent space, GENERALIST-based probabilities do not have epistasis.

To train the model, we write the total log-likelihood of the data:

$$\mathcal{L} = \sum_{n,l,a} \sigma_{anl} \log \pi_{anl} = -\sum_{n,l,a,k} \sigma_{anl} z_{nk} \theta_{akl} - \sum_{n,l} \log \Omega_{nl}. \tag{2}$$

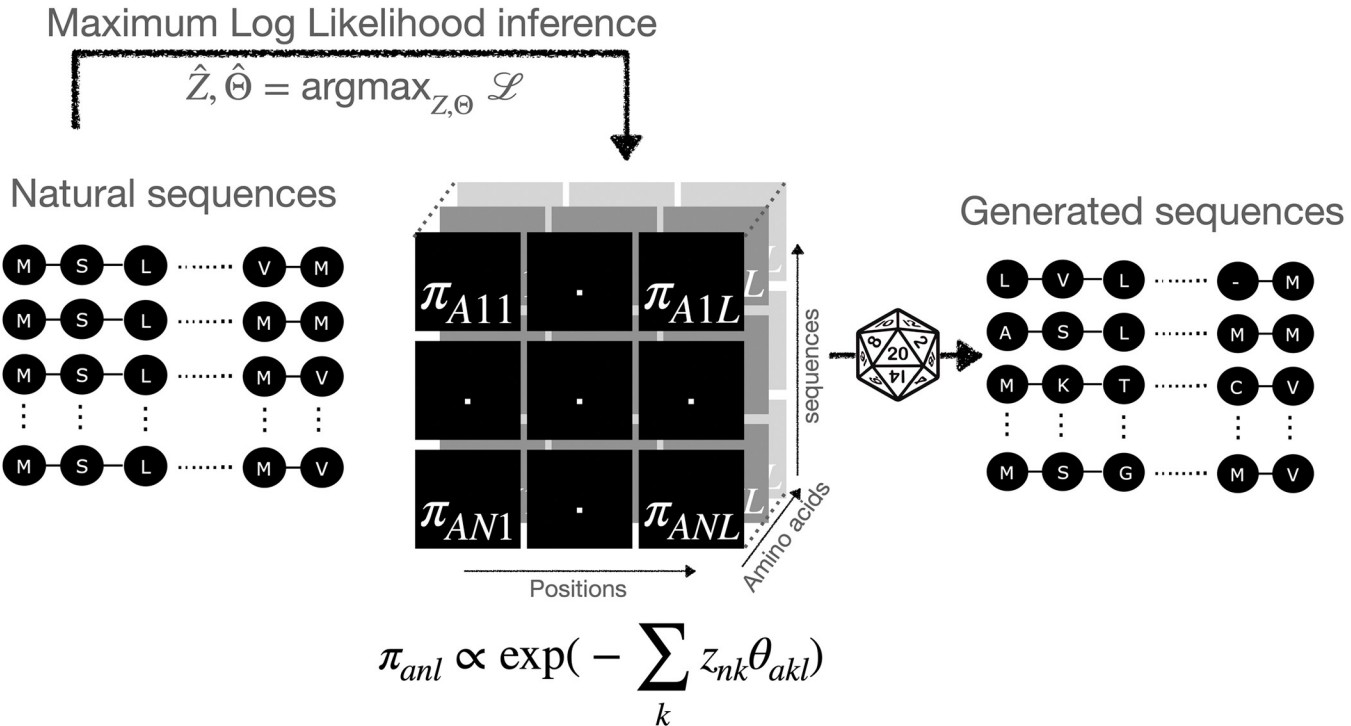

**Fig 1. Schematic of the GENERALIST approach.** Sequences are modeled as arising from sequence-specific Gibbs-Boltzmann distributions over categorical variables. The inferred probabilities are used to generate new sequences.

The gradients of the log-likelihood with respect to position- and amino-acid dependent parameters $\theta_{akl}$ and $z_{nk}$ are analytical (Section 1 in S1 Text). To infer the parameters as maximizers of the log-likelihood, we initialize them using uniformly distributed positive and negative values. We then simultaneously infer $z's$ and $\theta's$ using maximum likelihood inference (**Section 1 in S1Text** ). To generate *de novo* sequences, we randomly sample the learned latent variables, $z$, with replacement. The sampled $z's$, along with the energy parameters $\theta$ are used to define the probabilities in Eq (1), which are then used to generate sequences.

Below, we present our results for two proteins: Bovine Pancreatic Trypsin Inhibitor or BPTI, a small protein domain comprising 51 residues with a large MSA of 16569 sequences and epidermal growth factor receptor or EGFR, a large protein comprising 1091 residues with a small MSA of 1010 sequences. In S1 Text, we show our analyses for dihydrofolate reductase or DHFR (158 residues, 7164 sequences in the MSA), p53 (341 residues, 785 sequences in the MSA), and mammalian target of rapamycin or mTor (2549 residues, 520 sequences in the MSA). Details of model training can be found in Section 1 in S1 Text.

## Optimizing the latent space dimension in GENERALIST

GENERALIST is a latent space model. Increasing latent space dimension typically improves the ability of the generated ensemble to accurately capture summary statistics of the data (for example, amino acid frequencies and covariation). At the same time, a high dimensional latent space can result in a generated ensemble that is nearly identical to the natural one; trivially reproducing all statistics but failing to generate new sequences. Indeed, a common challenge with latent space models such as GENERALIST is selecting an appropriate complexity to avoid overfitting.

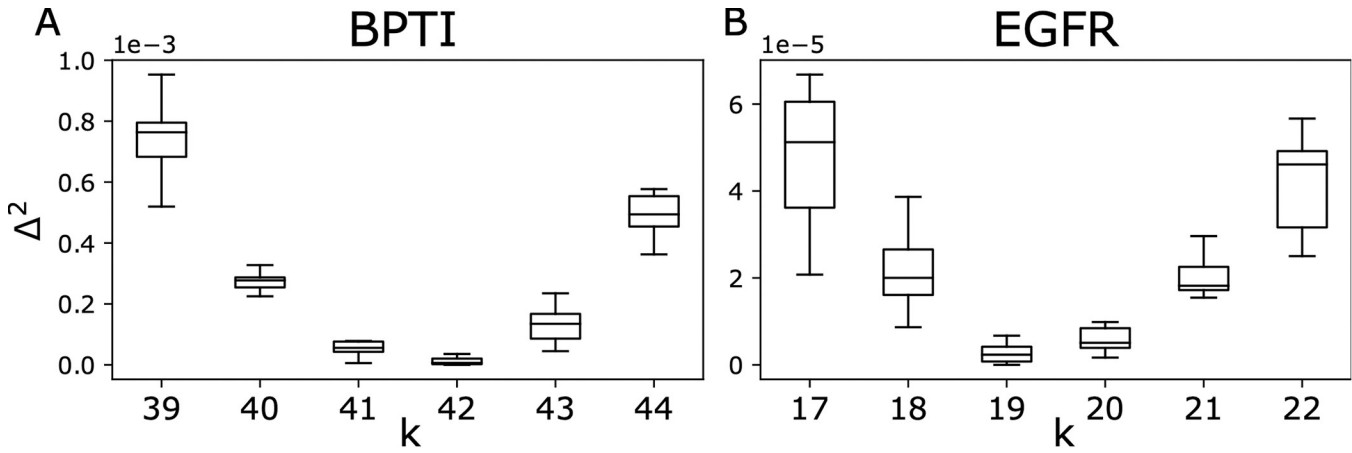

**Fig 2. Optimization for the latent dimension for BPTI (left panel) and EGFR (right panel).** The optimized value $\Delta^2 = (\langle H_{min}\rangle_{\text{generated}} - \langle H_{min}\rangle_{\text{natural}})^2$ (y-axis) is plotted against the latent dimension $K$ (x-axis). For each sequence in an ensemble (natural or generated) $H_{min}$ is calculated by obtaining the minimum fractional Hamming distance to the natural sequences. Each box plot represents 10 runs, each run with a different random initialization, for each latent dimension. The optimum latent dimension is determined as the one that minimizes the average value of $\Delta^2$ across the 10 runs. Optimal dimension for BPTI is 42, and for EGFR is 19.

In order to tune the latent dimension to accurately represent the data while avoiding overfitting, we compute the fractional Hamming distance from each generated sequence to all the natural sequences, this allows us to define the closest natural neighbor to a given generated sequence with the corresponding fractional Hamming distance $H_{min}$. $H_{min}$ for all the generated sequences defines the distribution of the minimum fractional Hamming distances between generated sequences and MSA. Similarly, we calculate the fractional Hamming distance between all natural sequences and their closest natural neighbor, which provides a distribution $H_{min}$ within MSA. The optimum latent dimension is then determined by matching the means these two distributions; we would like the generated ensemble to be on average as distant from its closest natural neighbors as the natural sequences are from their closest neighbor. Therefore, we define a difference $\Delta^2 = (\langle H_{min}\rangle_{\text{generated}} - \langle H_{min}\rangle_{\text{natural}})^2$. In Fig 2 (S1 Fig), we show this value reaches a minimum. When the latent dimension is smaller than the optimal one, the model is underfitted, with the $\langle H_{min}\rangle_{\text{generated}}$ greater than $\langle H_{min}\rangle_{\text{natural}}$. For a latent dimension greater than the optimum, the model is overfitted and $\langle H_{min}\rangle_{\text{generated}}$ gets smaller than $\langle H_{min}\rangle_{\text{natural}}$.

Since the log-likelihood is a non-convex function with several indeterminacies [21,22], different parameter initializations may result in different inferred parameters. To obtain robust performance, we train GENERALIST 10 times starting from arbitrary initializations (Section 1 in S1 Text). Notably, the performance of the model as measured by reproduction of sequence statistics is robust across multiple runs (S2 Fig). The optimum latent dimension $K$ is then determined as the one that minimizes $\Delta^2$ averaged across 10 runs. We find the optimum latent dimension for BPTI is $K = 42$, and for EGFR $K = 19$ using the average from the different runs. After deciding the latent dimension based on average of $\Delta^2$, we use the trained model with the smallest $\Delta^2$ for further analysis. We use the same metric to optimize for VAE and ArDCA hyperparameters (Section 4 in S1 Text and S3 and S4 Figs).

## Results

### GENERALIST reproduces high order summary statistics of natural sequences

A key metric to evaluate the accuracy of generative models is their ability to reproduce summary statistics of sequences (Section 3 in S1 Text). In Fig 3A and 3B, we show for BPTI and

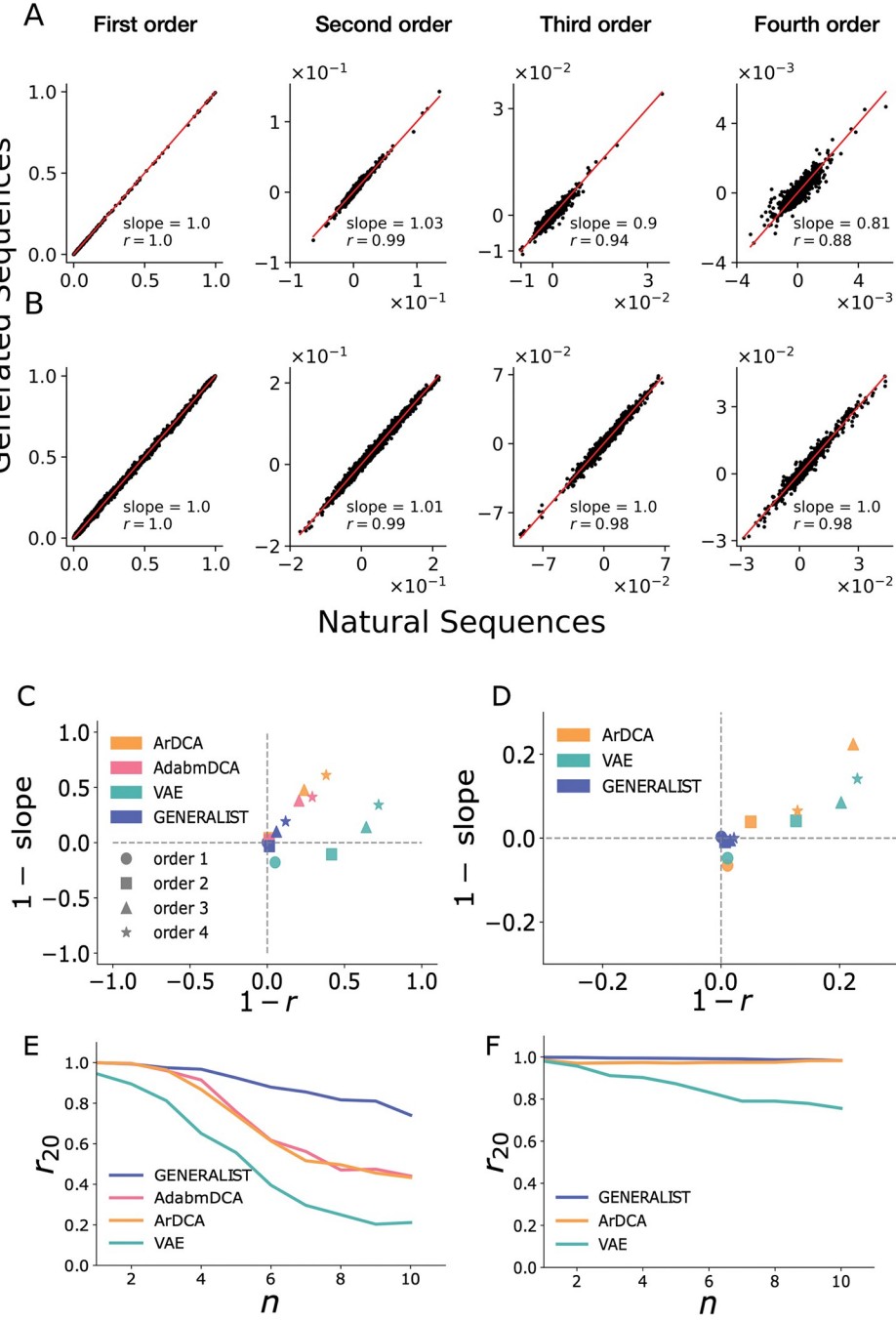

**Fig 3. Panels A and B.** Comparison of amino acid frequencies and cumulants up to order 4 calculated from GENERALIST-generated in silico ensembles (y-axis) and the natural sequences (x-axis) for BPTI (panel A) and EGFR (panel B). **Panels C and D.** 1 –Pearson correlation coefficient versus 1 –slope of the best fit line for the comparison between amino acid frequencies, and cumulants up to order 4 for GENERALIST, ArDCA, adabmDCA, and VAEs shown for BPTI (panel **C**) and EGFR (panel **D**). **Panels E and F.** The average Pearson correlation coefficient between frequencies of top 20 amino acid combinations of order n (x-axis) averaged across different combinations (y-axis) for GENERALIST, ArDCA, adabmDCA, and VAEs shown for BPTI (panel **E**) and EGFR (panel **F**).

EGFR that GENERALIST accurately reproduces amino acid frequencies and cumulants (central moments) up to order 4. Notably, as seen in Fig 3C and 3D (S7 Fig), while adabmDCA, ArDCA, and VAE-based predictions of positional frequency statistics correlate strongly with those observed in the natural sequences (Section 3 in S1 Text); these methods typically under-predict these statistics (quantified by the slope of the best fit line). To keep all models on an equal footing, we turned off phylogenetic corrections in adabmDCA and ArDCA. That is, we set the phylogenetic weights of all sequences to one.

Next, we investigated the ability of the generated ensembles to reproduce higher order summary statistics. Most amino acid combinations of order higher than 4 are rarely found in natural MSAs. We therefore used a recently introduced metric $r_{20}$ that measures the average Pearson correlation between the occurrence frequency of the top 20 amino acid combinations of any given order [25]. By computing the frequency of only the most frequent amino acid combinations, the $r_{20}$ metric is robust to sampling noise [25]. In Fig 3E and 3F (S8 Fig), we show that GENERALIST accurately captures co-occurrence frequencies of the most frequent amino acid combinations up to order 10. The ability of GENERALIST to capture these higher order statistics did not depend on restricting our attention to the top 20 amino acid combinations (S9 Fig). In comparison, adabmDCA, ArDCA, and VAEs led to less accurate predictions about higher order correlations when the MSAs were large (BPTI in the main text and DHFR in the SI). Importantly, the ensembles generated using VAEs did not exhibit a systematic trend toward more accurate predictions when the latent space dimension was increased unlike GENERALIST, where increasing the latent dimension $K$ corresponds to higher $r_{20}$. (S5 and S6 Figs).

These results conclusively show that GENERALIST-based sequence ensembles retain positional correlation information of arbitrarily high orders observed in naturally occurring sequences for large proteins as well as for proteins with very small MSAs. Table 1 provides a summary of the statistics results for all protein families studied here.

## GENERALIST finds conservative optimal sequences

A key feature of generative models is the ability to assign probabilities to arbitrary sequences and therefore find local sequence optima (sequences corresponding to the local maximum of the probability). The local optima inform us about the local structure of the inferred sequence space energy landscape and their relationship to naturally occurring sequences [9,10,15]. For example, if the generative models are purely data-driven, that is, if they do not incorporate any information about structure/function/fitness, it may be desirable that the local optima are in the vicinity of natural sequences.

To test the relationship between local optimum sequences and natural sequences, we use GENERALIST, adabmDCA, and ArDCA to obtain locally optimal sequences. VAE was not included because VAEs involve a nonlinear transformation from the latent space to the sequence space and therefore the probability in the sequence space is difficult to calculate.

**Table 1. Summary of GENERALIST performance on test protein families.** $r^1$ represents the Pearson coefficient of correlation between single site frequencies calculated from the generated ensemble and the natural MSA. $r^m$ where m>1, represents the Pearson coefficient of correlation between $m^{th}$ cumulants obtain from the generated sequences and the natural sequences.

| Protein | MSA Size | Sequence Length | Optimum K | $r^1$ | $r^2$ | $r^3$ | $r^4$ | $\Delta^2$ |
|---------|----------|-----------------|-----------|-------|-------|-------|-------|------------|
| BPTI | 16569 | 51 | 42 | 1 | 0.99 | 0.94 | 0.88 | $4.3 \times 10^{-6}$ |
| EGFR | 1010 | 1091 | 19 | 1 | 0.99 | 0.98 | 0.98 | $4.8 \times 10^{-7}$ |
| DHFR | 7164 | 158 | 64 | 1 | 0.99 | 0.95 | 0.95 | $2.3 \times 10^{-6}$ |
| P53 | 785 | 341 | 17 | 1 | 1 | 0.98 | 0.98 | $6.2 \times 10^{-6}$ |
| MTOR | 520 | 2549 | 12 | 1 | 1 | 0.98 | 0.98 | $1.0 \times 10^{-5}$ |

We obtained local minima in adabmDCA and ArDCA using a random search (Section 5 in S1 Text). Briefly, we start from sequences in the natural MSA and randomly mutated amino acids while only accepting mutations that improve sequence probability as evaluated by the model. Multiple iterations of this operation lead to local optimum sequences. The local optimum sequences predicted by GENERALIST were obtained by finding the highest probability sequence corresponding to the latent space embedding of natural sequences. The sequences obtained in this analysis are not a result of sampling a probability distribution but optimization of an inferred probability landscape. This analysis was only performed on BPTI where all three models could be trained in a reasonable time.

As seen in Fig 4A, adabmDCA generates locally optimal sequences that differed by a staggering 84% from the closest naturally occurring sequence neighbor. These optimal sequences were predicted to be significantly better compared to the starting natural sequences, with an average improvement by ~101 fold in probability at each position (Fig 4B, measured by log odds ratio). These local minima in the Potts model that do not resemble any natural sequences are reminiscent of the unwanted spurious minima in Hopfield networks [26]. Compared to adabmDCA, ArDCA generated local optimal sequences that were significantly more conservative (on an average, 17% difference compared to 84%) (Fig 4A). The optimal sequences were also predicted to be a relatively modest improvement over the starting natural sequence with an improvement by ~1.5 fold in probability at each position (Fig 4B). Like ArDCA, GENERALIST-based local optima were significantly more conservative. As seen in Fig 4A, the local optimum sequences differed from the closest naturally occurring sequences on an average by 8%. As seen in Fig 4B, the per amino acid improvement was only ~1.1 fold.

To test whether these sequences potentially fold in stable 3D structures, we used Alpha-Fold2 [27], a recent machine learning method that can predict 3D structures from sequences and MSAs (Section 6 in S1 Text). We used the sequence-averaged predicted local distance difference test (plddt) as a proxy for quality of predicted structures. The plddt score is a computationally predicted local distance difference test [27] and measures the confidence of the AlphaFold2 algorithm in the predicted structure. Previous studies have shown that a sequence average plddt of >80 corresponds to sequences that are likely to fold in stable 3D structures [28]. In contrast, a lower plddt score may either imply an unstructured (disordered) protein (or region of a protein) or a low algorithmic confidence in the predicted structure. Given that wild type BPTI is a highly structured protein, it is reasonable to assume that a low plddt score is likely to result in non-functional proteins. As seen in Fig 4C, local optimal sequences imputed by adabmDCA were predicted to be significantly worse folders compared to both

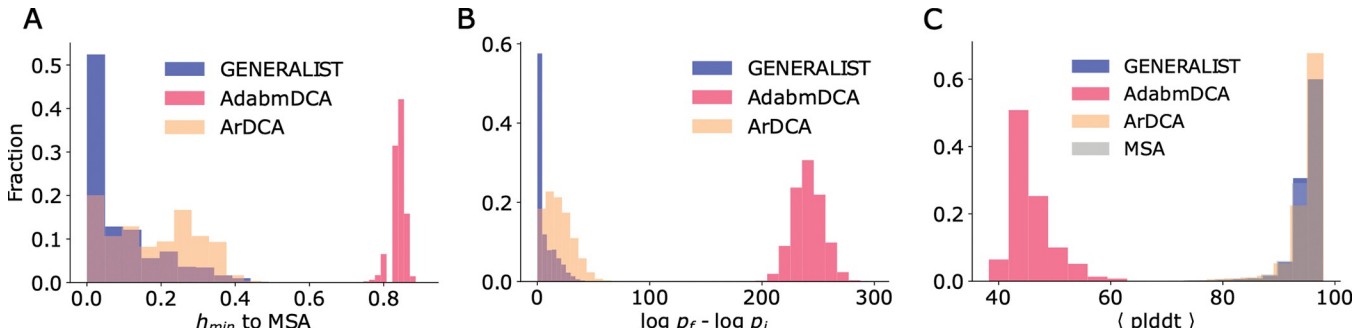

**Fig 4. Panel A.** The distribution of distances to the nearest natural neighbor from sequences optimized using GENERALIST, ArDCA, and adabmDCA modeled probabilities. **Panel B.** The log-fold improvement in probabilities between the starting sequence and the local optimum. **Panel C.** Sequence-averaged plddt scores for AlphaFold2 predicted structures for the locally optimum sequences for all models, and the starting natural sequences labeled "MSA".

GENERALIST and ArDCA. While ArDCA and GENERALIST produce sequences that were predicted to be comparable by AlphaFold2 on average.

These results show that GENERALIST can identify local optima in the sequence space. These optima are typically not seen in nature. The optimal sequences predicted by GENERALIST were also predicted by AlphaFold2 to fold in stable 3D structures.

## GENERALIST reproduces the distributions sequence distances from natural ensembles

In addition to reproducing the summary statistics (Fig 3), an important test for generative models is capturing various distribution of inter-sequence distances in the natural ensemble [8,29]. To that end, we evaluated three different statistics for all generated ensembles: (a) the distribution of distances between pairs of randomly picked sequences, (b) the distribution of nearest neighbor distances within an ensemble, and (c) the distribution of distances to the nearest natural neighbor.

In Fig 5A and 5B (S10 Fig), we plot the distribution of fractional Hamming distances between pairs of random sequences in an ensemble. We see that all generative models, except for the VAEs for BPTI (Fig 5A) and DHFR (S10 Fig), accurately reproduced this distribution, implying that most ensembles captured the expanse of the natural sequence ensemble. Ensembles generated using VAEs comprised sequences that are on average are closer to each other than natural sequences.

The distribution of nearest neighbor distances portrayed a more complex picture. In Fig 5C and 5D (S10 Fig and Section 2 in S1 Text), we plot the distribution of fractional Hamming

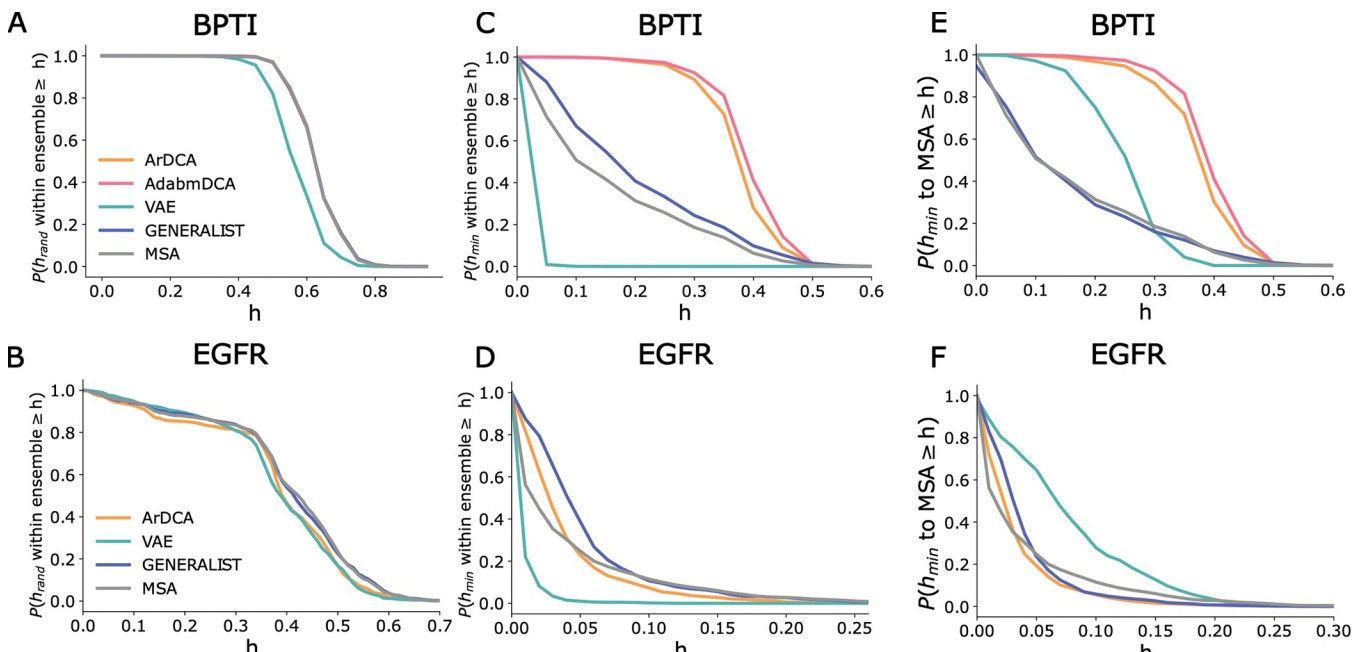

**Fig 5. Panels A and B.** Distribution of fractional Hamming distances between random pairs of sequences within an ensemble shown as the fraction of pairs for which the hamming distance $h_{rand}$ within ensemble (y-axis) is greater or equal than value $h$ (x-axis). Panel A: BPTI, Panel B: EGFR. **Panels C and D.** Distribution of fractional Hamming distances to the closest sequence within an ensemble for different models shown as the fraction of sequences for which the minimum hamming distance $h_{min}$ within ensemble (y-axis) is greater or equal than value $h$ (x-axis). Panel C: BPTI, Pabel D: EGFR. **Panels E and F.** Distribution of fractional Hamming distances to closest natural sequence for different models shown as the fraction of sequences for which the minimum hamming distance $h_{min}$ to MSA (y-axis) is greater or equal than value $h$ (x-axis). Panel E: BPTI, Panel F: EGFR.

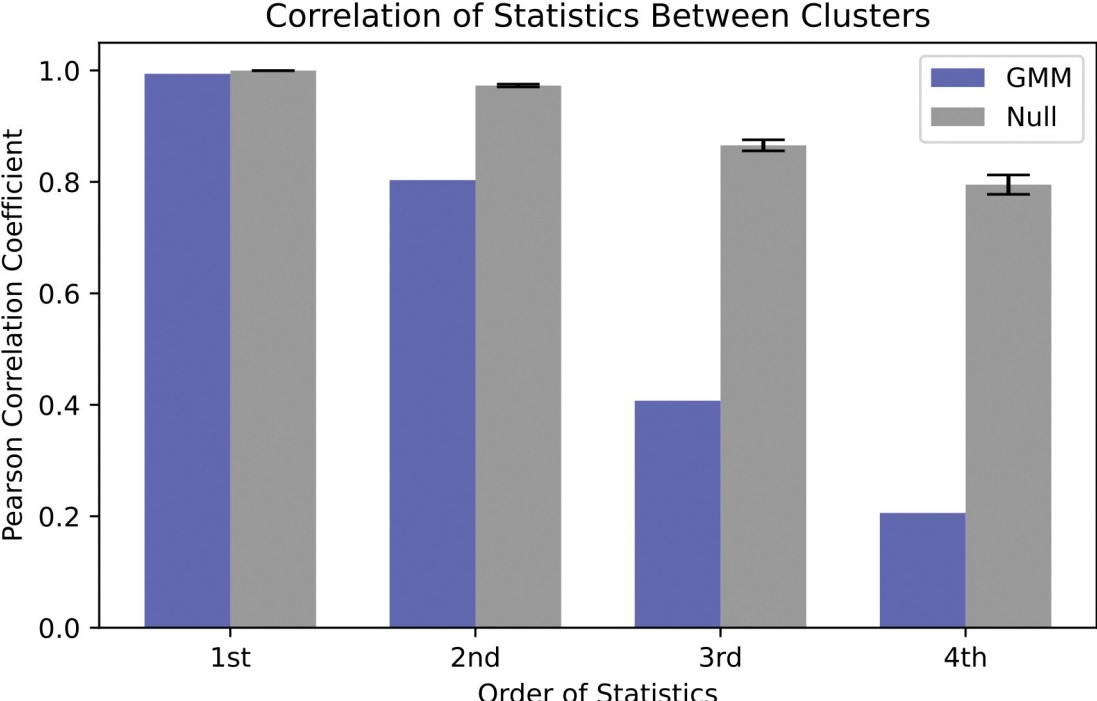

**Fig 6. Statistics of the cluster assignments for BPTI.** Pearson correlation coefficient between higher order statistics of amino acid occurrences in sequences belonging to the two clusters. Cluster sizes are 12646 and 3923.

distances to the nearest neighbor within the ensemble. When MSA was large (BPTI in Fig 5C and DHFR in S10 Fig), ArDCA/adabmDCA generated ensemble comprised sequences that were farther away from their ensemble nearest neighbors compared to natural sequences. In contrast, for the smallest MSA (mTor in S10 Fig), ArDCA generated ensembles comprised sequences with closer nearest neighbors compared to natural sequences. For proteins P53 and EGFR, ArDCA model was able to the capture the distance distribution (Figs 5D and S10). VAEs generated ensemble always comprised sequences that were on average closer to their nearest neighbors than natural sequences. Finally, GENERALIST generated ensemble closely reproduced the density of nearest neighbor sequences observed and shows robustness to MSA size.

Next, we compared the distance distribution to the nearest natural neighbor. Here too, GENERALIST generated ensembles closely reproduced the density of nearest neighbor sequences (Figs 5E, 5F, and S10). In contrast, ArDCA/adabmDCA generated sequences were farther from the natural sequences if the MSA was large (BPTI in Fig 5E and DHFR in S10 Fig). While ArDCA shows accurate representation of the nearest natural neighbor distance distribution for EGFR (Fig 5B) and P53 (S10 Fig), it shows overfitting to the smallest MSA of mTor (S10 Fig). Finally, VAEs generated ensemble comprised sequences that were farther away compared to natural sequences for all proteins.

These results show that an optimally tuned GENERALIST ensemble can capture various aspects of the density of sequences in the natural ensemble.

### The latent space representation identifies sequence clusters with differing positional covariance structure

One significant advantage of GENERALIST is that its latent space representation embeds individual sequences in a smaller dimensional latent space. Notably, the covariance structure in

the latent space (covariance of different latent dimensions across samples) is directly analogous to statistical relationships among the amino acid positions. Consequently, if the latent space partition into several sub-clusters of differing covariance, the corresponding sequences are similarly grouped, each cluster characterized by distinct statistical properties. To test this, we used Gaussian mixture modeling (GMM) to identify clusters in the latent space for BPTI (Section 7 in S1 Text). We identified 2 clusters in the data (S11 Fig) and found that they captured distinct higher order statistical characteristics. To show this, we computed the Pearson correlation between amino acid frequencies and covariances of multiple orders between sequences belonging to the two clusters. These statistics were compared to a null model where the sequences were partitioned randomly using the same cluster sizes as obtained using the GMMs. As seen in Fig 6, comparison between GMM clusters and the null clusters show that the latent space can be partitioned into regions representing different statistics. This shows that GENERALIST can be used to identify clusters of sequences with unique statistical structure.

## Discussion

Generative models of protein sequence families are an important tool for protein scientists and engineers alike. Ideally, these models should be simple to learn, tunable, and accurate, especially when studying proteins of significant clinical interest which tend to be large proteins with small MSAs.

In this work, we examined three state-of-the-art models. Potts models could only be used to model small sequences, limiting their application to single domains and small proteins. Moreover, these models could not be tuned. The sequence ensemble generated by Potts models could not reproduce the density of sequences in the natural ensemble and had optima that appeared unnatural. In contrast, the autoregressive DCA model was significantly more efficient in model fitting and more accurate in reproducing summary statistics such as frequencies of higher order amino acid combinations (Fig 3). The model also reproduced local optima that were computationally deemed to fold in stable 3D structures (Fig 4). However, given that the model has $O(L^2)$ parameters for proteins of sequence length $L$, this model overfits the training data when modeling human proteins of significant clinical interest which are large and have small MSAs (mTOR S10 Fig).

Neural networks based variational autoencoders were efficient and did not appear to overfit the training data (Fig 5). Overall, these models were less accurate in predicting summary statistics of sequences compared to GENERALIST, the Potts model, and the autoregressive DCA model. Consistently VAE generated sequences that are much closer to each other than the natural sequences, indicating a separation in the sequences space between the generated and the natural ensemble. At the same time, potentially owing to model complexity (and therefore parameter non-identifiability) and low amounts of training data, the models appeared to not have any systematic trends with respect to accuracy and overfit as a function of the dimension of the latent space.

In contrast, GENERALIST is accurate and tunable, allowing us to analyze large proteins with small MSAs. Notably, given its simple structure, there are several avenues of improving GENERALIST. For example, function/fitness information obtained from deep mutational scanning can be incorporated as constraints on the energies and phylogenetic information can be imposed as constraints on the latent space. Finally, GENERALIST can be easily reformulated for any other categorical data, for example, presence/absence of single nucleotide polymorphisms or nucleotide sequences. We believe that GENERALIST will be an asset for protein scientists and engineers alike.

## Supporting information

**S1 Fig. Optimization for the latent dimension.** The optimized value is $\Delta^2 = (\langle H_{min}$ from generated ensemble to MSA$\rangle - \langle H_{min}$ within MSA$\rangle)^2$. Each box plot represents 10 runs for each latent dimension. The optimum latent dimension for DHFR is 64, for P53 is 17 and for and for MTOR is 12.
(TIF)

**S2 Fig. Box plot of Pearson coefficient of correlation obtained from different initializations for the model.** Pearson coefficient of correlation $r$ (y-axis) generated from calculating the cumulants of the generated ensemble and the natural ensemble vs order of cumulants $n$ (x-axis). Each box represents 10 runs for the same latent dimension of GENERALIST.
(TIF)

**S3 Fig. Optimization for ArDCA hyperparameters.** The regularization for the couplings and the field, $\lambda_J$ and $\lambda_H$ respectively. The optimized value is $\Delta^2 = (\langle H_{min}$ from generated ensemble to MSA$\rangle - \langle H_{min}$ within MSA$\rangle)^2$.
(TIF)

**S4 Fig. Optimization for VAE latent dimension.** The optimized value is $\Delta^2 = (\langle H_{min}$ from generated ensemble to MSA$\rangle - \langle H_{min}$ within MSA$\rangle)^2$.
(TIF)

**S5 Fig. $r_{20}$ of different latent dimensions of GENERALIST.** The average Pearson correlation coefficient between frequencies of top 20 amino acid combination of order n (x-axis) averaged across different combinations (y-axis) for GENERALIST model trained with different latent dimensions $K$ (legend). Each subplot represents a different protein from left to right: BPTI, DHFR, P53, EGFR, MTOR.
(TIF)

**S6 Fig. $r_{20}$ of different latent dimensions of VAE.** The average Pearson correlation coefficient between frequencies of top 20 amino acid combination of order n (x-axis) averaged across different combinations (y-axis) for VAE model trained with different latent dimensions $K$ (legend). Each subplot represents a different protein from left to right: BPTI, DHFR, P53, EGFR, MTOR.
(TIF)

**S7 Fig. Comparing the accuracy of different models in capturing statistics of different orders through slope and Pearson correlation.** For each order of statistics (grey legend panel A), the frequencies of different amino-acid strings are obtained from Natural and generated ensemble. For order $>2$, mean removed frequencies are used. The slope of the best fit line of those frequencies as well as the Pearson coefficient of correlation is obtained. 1—slope (y-axis) vs 1—Pearson correlation (x-axis) is plotted for different models (legend).
(TIF)

**S8 Fig. $r_{20}$ comparison between different models for DHFR, P53 and MTOR.** The average Pearson correlation coefficient between frequencies of top 20 amino acid combinations of order $n$ (x-axis) averaged across different combinations (y-axis). **Panel A,** protein DHFR, **panel B** protein P53 and **panel C** is protein MTOR.
(TIF)

**S9 Fig. Using top 10 and 50 frequencies for average Person Correlation for BPTI and EGFR. Panels A, B.** The average Pearson correlation coefficient between frequencies of the

top 10 amino acid combinations of order n (x-axis) averaged across different combinations (y-axis). **Panels C, D** Same as A and B but using the frequencies of the top 50 amino acid combinations to obtain the average Pearson correlation.
(TIF)

**S10 Fig. Comparison of the distribution of fractional Hamming distances between the generated and natural sequences for DHFR, P53 and MTOR. Panels A, B and C.** Distribution of fractional Hamming distances between random pairs of sequences within an ensemble shown as the fraction of pairs for which the hamming distance $h_{rand}$ (y-axis) is greater or equal than value $h$ (x-axis). **Panels D, E and F**. Distribution of fractional Hamming distances to the closest sequence within an ensemble for different models shown as the fraction of sequences for which the minimum hamming distance $h_{\min}$(y-axis) is greater or equal than value $h$ (x-axis). **Panels G, H and I**. Distribution of fractional Hamming distances to closest natural sequence for different models shown as the fraction of sequences for which the minimum hamming distance $h_{\min}$ to MSA (y-axis) is greater or equal than value $h$ (x-axis).
(TIF)

**S11 Fig. Optimization for the number of Gaussians in GMM.** The optimal value is N = 2. This was chosen with the Jaccard Index, a measure of similarity between cluster assignments. Each box plots represents the Jaccard Index over 20 iterations of comparing the assigned clusters of two GMMs for the labeled number of Gaussians.
(TIF)

**S1 Text. Details of the implementation of GENERALIST, the different performance metrics as well as the models used for benchmarking.**
(PDF)

## Acknowledgments

We would like to thank Juannan Zhou for critical comments and useful discussions.

## Author Contributions

**Conceptualization:** Hoda Akl, Purushottam D. Dixit.

**Data curation:** Hoda Akl, Brooke Emison, Purushottam D. Dixit.

**Formal analysis:** Hoda Akl, Brooke Emison, Xiaochuan Zhao, Arup Mondal.

**Funding acquisition:** Purushottam D. Dixit.

**Software:** Hoda Akl, Brooke Emison.

**Supervision:** Alberto Perez, Purushottam D. Dixit.

**Writing – original draft:** Hoda Akl, Brooke Emison.

**Writing – review & editing:** Hoda Akl, Brooke Emison.

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
