## [Decision Letter · Decision Letter 0]

22 Mar 2023

Dear Dr. Dixit,

Thank you very much for submitting your manuscript "GENERALIST: An efficient generative model for protein sequence families" for consideration at PLOS Computational Biology.

As with all papers reviewed by the journal, your manuscript was reviewed by members of the editorial board and by several independent reviewers. In light of the reviews (below this email), we would like to invite the resubmission of a significantly-revised version that takes into account the reviewers' comments.

The reviewers have clearly approciated the novelty and potential interest of the proposed approach, but also raised some concerns in particular with respect to the motivation of the current approach, comparison to existing models, overfitting with respect to phylogeny and some not fully justified claims in the manuscript.

We cannot make any decision about publication until we have seen the revised manuscript and your response to the reviewers' comments. Your revised manuscript is also likely to be sent to reviewers for further evaluation.

Sincerely,

Martin Weigt

Guest Editor

PLOS Computational Biology

Arne Elofsson

Section Editor

PLOS Computational Biology

The reviewers have clearly approciated the novelty and potential interest of the proposed approach, but also raised some concerns in particular with respect to the motivation of the current approach, comparison to existing models, overfitting with respect to phylogeny and some not fully justified claims in the manuscript.

Reviewer's Responses to Questions

**Comments to the Authors:**

Reviewer #1: In this paper, Akl and collaborators describe a new generative model framework for describing properties of protein families. Their approach uses a set of latent variables, the number of which is chosen by comparisons between the statistics of the models that they infer and the real data, to quantify the probability for particular sequences. They apply their method to data sets for several proteins. Here they show that their method is able to reproduce statistics of the data that in some ways appears to exceed other, alternative approaches.

The method used by this paper is novel, and new contributions in this area are welcome. However, the ideas presented here are in some cases poorly explored. Some characterizations of past work also seem incomplete. In addition, I have technical questions about the work. My detailed comments are below:

1. Justification for the new approach. In multiple places, the authors attempt to justify their approach with arguments about computational efficiency or scaling. For example, lines 72-74, referring to Potts models: “The associated numerical inference is computationally inefficient, limiting their application to small proteins and protein domains ( ∼ 100 residues).” There are also claims about large numbers of hyperparameters for Potts models. There are several problems with these justifications for the current work.

First, the claims above are not generically true. Potts models have been trained using a variety of algorithms for large proteins. This includes work by (Hopf et al, Nat Biotechnol 2017) using pseudolikelihood, or (Louie et al, PNAS 2018) using a method known as minimum probability flow, among others. Indeed, the authors themselves use other algorithms for inferring Potts models in this very paper on proteins with L > 2000. While some methods may employ multiple hyperparameters, most really only use a couple of regularization strengths (one for “fields” and one for “couplings”) and perhaps a parameter for data compression. This is not very different from the authors’ proposed method, where the number of latent variables is unknown and must be estimated.

Second, despite the claim that computational efficiency is important, the paper never presents any comparison of run time or other computational parameters for the various methods that they test. This is surprising, and it makes claims about “efficiency” impossible to evaluate in a quantitative way.

Finally, and perhaps most importantly, the justification of computational efficiency doesn’t seem necessary at all. Why not simply motivate this work by emphasizing the latent variable approach, which differs from most methods in this area, and the statistical properties of the model?

2. Quantitative comparisons with other methods. In Figure 3, if my understanding of the SI is correct, then the higher order correlations presented are not actually connected correlations (except for the second order ones). Instead, the expected “mean,” given by the product of individual AA frequencies, is subtracted from the frequency in simulations. In this case, the “higher order” terms can actually depend substantially on lower order moments. To truly get independent statistics, one should instead use the connected correlations or cumulants. More generally, for statistics that involve large numbers of sites/residues, it would be helpful for the authors to justify why one should believe that these statistics are meaningful in the face of finite sampling noise. Surely, correlations for combinations of ~ 10 residues will be woefully undersampled in any data set of realistic size, for example. Simulations could help the authors to explore these issues.

In Figure 4, the authors compare several aggregate statistics for different models. Figures 4A-B seem to suggest that sequences from GENERALIST are typically highly similar to ones from the MSA and inferred “optimal” sequences. Is it clear that GENERALIST is significantly exploring the landscape of possible sequences?

In addition, it would be helpful for the authors to more thoroughly explain the plddt score, how this is calculated, and the motivation behind its use. How is a plddt score calculated for a particular sequence (natural or artificial)? In principle, sequences with some disorder could also be functional. Is it clear that larger plddt scores are better?

While statistics of the sequence ensemble are interesting, these may be primarily of specialized interest. In order to make a full comparison with alternative methods, it would be useful to explore how well GENERALIST is able to predict the functional effects of mutations, for example.

Some analyses suggest overfitting of Potts models to particular data sets, but I would be inclined to argue that this is not an entirely fair comparison. The authors fine-tune their own analyses, selecting a number of latent factors that is appropriate for each data set. In contrast, default parameters are used for the different DCA model-fitting methods. One might expect that overfitting could easily be countered by appropriately adjusting regularization in the Potts model methods.

3. Unique contributions of this approach. The authors have developed a detailed method to infer latent factors describing ensembles of protein sequences. Remarkably, however, the potential insights from the latent factors themselves appear to be completely unexplored. Do these factors have any clear biological interpretation? Could they be used for categorizing sequences in any way? Or are these latent factors purely for computational expedience?

4. Model fitting. It would be helpful for the authors to more clearly explain the methods that they have used for fitting their model. Based on eq. (2), one must compute the partition function (or complicated averages) to fully evaluate the derivative. In particular, the pi functions appear to be probabilities within the model rather than in the data, so it’s not obvious how to obtain these without either estimating them through simulation or computing the partition function.

Optimization with latent variables is generally non-convex, making this problem complex. How sensitive are the results to initial conditions or any approaches for regularization? Are latent factors similar between runs, or do they appear to be different each time? How does general performance vary between instances?

Finally, sampling from the model could be more completely explained. Once the z’s have been learned, how does one resample the z’s? Are these assumed to come from a distribution?

5. References. There are several places where the authors could augment or correct references in the current version of the text. In reference to the effects of mutations on fitness, (Ferguson et al, Immunity 2013) precedes ref. 8. (Figliuzzi et al, Mol Biol Evol 2015) extended this concept to protein families. And of course, related ideas have now been applied in many subsequent papers. In line 89, it may be best to refer to (Riesselman et al, Nat Methods 2018) and subsequent work. Regarding the identification of “optimal sequences,” to my knowledge this was first performed in (Barton et al, PNAS 2015), which also provided a biological interpretation for the optimal sequences.

More generally, this field has greatly expanded and deepened in the last ~10 years. It would be helpful for the authors to expand the present references to highlight original research in addition to recent reviews and to point to additional relevant research in this area.

Minor comments:

The notation for indices used here is a little bit confusing. What is the difference between the position in the protein sequence “l” and the index “n”? Without further explanation it seems like these two coordinates are indexing the same thing, and it’s unclear why both are needed. This is made more clear in the SI, but this should be clarified in the main text.

Reviewer #2: The review is uploaded as an attachment

Reviewer #3: In this manuscript, the authors propose a generative model, called GENERALIST, to model natural protein multiple sequence alignments (MSA). The model is very simple and this makes it extremely appealing. It consists in constructing K "profile models", interpreted as K features that describe the sequences. Each natural sequence is then represented as a linear superposition of these K profile models, with sequence-dependent weights Z.

Once the parameters are inferred, one can extract the weights Z corresponding to one of the natural sequences, and sample new sequences according to the profile models associated with it. Because of the profile structure, each aminoacid in the sequence is sampled independently, which makes sampling easy and efficient.

The number K of features is chosen in such a way as to match the probability distribution of the sequence identity with the closest natural sequence in the MSA for (a) sequences generated from the model and (b) natural sequences.

The authors then test their model on 5 different natural MSA, and compare it with other state-of-the-art approaches such as Boltzmann machines (adabmDCA), autoregressive models (arDCA), and variational autoencoders (VAE). They claim that GENERALIST outperform these models. The claim is based on a series of observations:

1) Higher-order statistics of the natural MSA is better reproduced (Fig.3)

2) Local maxima of the model probability are closer to natural sequences and AlphaFold2 predicts them to fold better into the natural structure associated to the family (Fig.4).

3) GENERALIST closely reproduces the distributions of distances between pairs in the generated vs natural ensemble, of minimum distances within the ensemble and with the natural MSA.

Given these facts, my assessment of the paper is the following. I think that the method is definitely of interest and as such I would recommend publication on PLOS computational biology. However, I am not fully convinced that GENERALIST outperforms existing models and I would thus encourage the authors to tune down some statements that I find a bit too bold.

The reason is the following. Models such as adabmDCA, arDCA and VAE assume that sequences in the natural MSA are identically and independently generated from an unknown probability distribution, which is learned by the inference method. On the contrary, GENERALIST tries to learn a model of the whole MSA. In doing so, GENERALIST incurs the risk of overfitting the phylogeny that is always present in natural MSAs; while the other methods try to get rid of it (perhaps without much success).

In this sense, it is not surprising that GENERALIST does a better job in fitting the distance relationships of the natural MSA and the high-order statistics. Methods such as adabmDCA, by construction, try to get rid of the low-distance tail that characterizes the natural MSA. I am therefore a bit afraid that GENERALIST might be overfitting the phylogeny a bit. Whether this is a problem or not depends probably on the task that one is considering, and therefore GENERALIST might well outperform the other approaches in some applications. But I would encourage the authors to be a bit more cautious in the absence of an experimental comparison of the methods on a given task that is relevant for biology.

A minor point: in the abstract and introduction, “density of sequences” is not clear. I guess it should be “probability density of sequence distances”.

**Have the authors made all data and (if applicable) computational code underlying the findings in their manuscript fully available?**

Reviewer #1: Yes

Reviewer #2: Yes

Reviewer #3: Yes

PLOS authors have the option to publish the peer review history of their article (what does this mean?). If published, this will include your full peer review and any attached files.

Reviewer #1: No

Reviewer #2: No

Reviewer #3: No
---

## [Decision Letter · Decision Letter 1]

18 Jul 2023

Dear Dr. Dixit,

Thank you very much for submitting your manuscript "GENERALIST: A latent space based generative model for protein sequence families" for consideration at PLOS Computational Biology. As with all papers reviewed by the journal, your manuscript was reviewed by members of the editorial board and by several independent reviewers. The reviewers appreciated the attention to an important topic. Based on the reviews, we are likely to accept this manuscript for publication, providing that you modify the manuscript according to the review recommendations.

While all three reviewers appreciate the effort made by the authors in their revision, some smaller points remain still open, and some claims regarding previously published literature seem inaccurate and need further clarification.

Sincerely,

Martin Weigt

Guest Editor

PLOS Computational Biology

Arne Elofsson

Section Editor

PLOS Computational Biology

While all three reviewers appreciate the effort made by the authors in their revision, some smaller points remain still open, and some claims regarding previously published literature seem inaccurate and need further clarification.

Reviewer's Responses to Questions

**Comments to the Authors:**

Reviewer #1: I thank the reviewers for their responses, which have improved the paper. However, I must push back on a few points where the responses are inaccurate. In addition, one finding from the response raises some concerns.

1. In response to Reviewer 1 Comment 1: “However, while these approximate methods can potentially predict the effect of mutations and physical contacts between residues, they perform poorly when it comes to reproducing statistics of real sequences, which was the chief goal of our generative models and also of our work. The inaccuracy of approximate inference methods (pseudolikelihood, mean field, adaptive cluster expansion, etc.) in reproducing statistics of natural sequences was extensively explored by Figliuzzi et al. [1].”

The statement on the inaccuracy of alternative inference methods is not correct, as the authors can verify by consulting this reference (Figliuzzi et al., MBE 2018). In Figliuzzi et al., it is shown that pseudolikelihood does not do a good job of reproducing the statistics of the data. It is well-known that mean field approaches also do not reproduce statistics of the data. However, Boltzmann machine learning works very well (Figliuzzi et al., MBE 2018). Similarly, (Barton et al., Bioinformatics 2016) show good reproduction of statistics of the data using the adaptive cluster expansion method. The same is also true for ArDCA and minimum probability flow.

Continuing on, “We use the term “Potts models” to refer to a generalization of spin-glass models for categorical data wherein the “Hamiltonian” of any sequence is given by [Eq. R1]. […] We note that in our manuscript, we do not train Potts models for proteins with L > 2000. In fact, we only train the Potts model for BPT1 with L ~ 60 residues. What the reviewer may be referring to is our implementation of a previously developed autoregressive model (ArDCA) [2] which has a very similar probability distribution as the Potts model yet cannot be represented using a Hamiltonian in Eq. R1.”

This is splitting hairs, and by this same standard the model (or, if we are being very precise, the ensemble of models) inferred by pseudolikelihood is also not a Potts model.

In response to the number of hyperparameters used, the response to reviewers states that “We do not include the dimension of the latent space as a hyperparameter since its value can be computed from the data.” However, in the paper the authors state “The latent space dimension is tuned to achieve a user-desired tradeoff between the novelty of

generated sequences and the accuracy of the ensemble in reproducing properties of the natural MSAs.” (lines 112-115). So, the dimension is not a hyperparameter, but the cutoff used to determine the dimension is. One can compare GENERALIST with alternatives such as ArDCA and find a very similar number of hyperparameters.

Broadly, I have two recommendations. The first, specific recommendation would be to ensure that claims about previously developed methods in the literature are accurate. Secondly, I would suggest that the authors either motivate their work on its own merits – and indeed, their approach is interesting on its own – or substantiate claims about performance with data in this paper.

2. In response to Reviewer 1 Comment 4 and related comments from Reviewers 2 and 3, “Parameter inference in GENERALIST is indeed nonconvex with potential to get stuck in local optima. Based on the comments of this reviewer and others, we now alleviate this issue by training the GENERALIST model multiple times and choosing the one that has the highest likelihood. The latent factors are not similar between different runs, due to a variety of previously documented indeterminacies [10,11].”

This finding seems concerning for the interpretation of the model, if not its fit to the data. Are other features of the model, such as local sequence optima or sequence clusters, conserved between different runs? Of course, if a particular parameter initialization turns out to be a bad one and the fit is poor, then such an instance could be discarded. Here, I am asking whether or not similarly good fits to the data return similar insights.

The authors should note whether or not these findings are consistent, and if not, this point should at least be acknowledged and explained in the Discussion.

Reviewer #2: I thank the authors for their work on revising the manuscript. My comments have been addessed.

Reviewer #3: I read with great interest the authors' answers to the referees. Both the questions and the answers were very insightful. I confirm my previous assessment of the paper and suggest acceptance on PLOS Computational Biology.

However, I would like the authors to consider the following comments, based on their answers to the other referees. Lines below refer to the authors' answers to the referees:

1. (Lines 125-126) The authors have removed claims of computational efficiency. Why? I would suggest adding a table with the training and sampling times for the various models, which will be very useful for future reference.

2. (Lines 181-182) The authors mention the presence of spurious optima in bmDCA. I am not totally sure that the presence of local optima that are distinct from the natural sequences is a bad sign. On the contrary, it could be a sign that bmDCA is able to generalize. The only way to know this is to perform experimental tests of these sequences.

3. (Lines 222-229) The fact that GENERALIST does not account for epistasis in important. It should be mentioned explicitly in the main text.

4. (Lines 436-437) Are the authors sure that repeating the training 10 times is enough to solve the problem? (Did they check it in the specific example proposed by referee 2?) What would be a good criterion to decide how many times the training should be repeated?

5. (Lines 407-409) When talking about RBMs, it should be noted that RBMs have been applied to protein families in a series of works mostly by the group of Cocco and Monasson, e.g. "Learning protein constitutive motifs from sequence data", eLife, https://doi.org/10.7554/eLife.39397.001

A more direct comparison with these approaches, most notably comparing the latent space representations, would be nice, but I understand that it goes beyond the scope of this work. However, perhaps some of these works could be cited. (I wish to make clear that I am not an author in any of these papers, so there is no conflict of interest here.)

6. On the question of phylogeny, which was my most important concern and was shared by the other referees, I am not totally satisfied by the answer. I remain of the opinion that the other approaches work under different assumptions about phylogeny, such that a direct comparison might be misleading. However, I agree with the authors that this is one of the most important issues in generative modeling of proteins, and I think that GENERALIST will contribute a new tool to tackle it, so I accept the authors' point of view on this and do not require any change to the paper.

**Have the authors made all data and (if applicable) computational code underlying the findings in their manuscript fully available?**

Reviewer #1: Yes

Reviewer #2: Yes

Reviewer #3: Yes

PLOS authors have the option to publish the peer review history of their article (what does this mean?). If published, this will include your full peer review and any attached files.

Reviewer #1: No

Reviewer #2: No

Reviewer #3: No

Figure Files:

Data Requirements:

Reproducibility:

References:

---

## [Decision Letter · Decision Letter 2]

3 Nov 2023

Dear Dr. Dixit,

We are pleased to inform you that your manuscript 'GENERALIST: A latent space based generative model for protein sequence families' has been provisionally accepted for publication in PLOS Computational Biology.

Best regards,

Martin Weigt

Guest Editor

PLOS Computational Biology

Arne Elofsson

Section Editor

PLOS Computational Biology

Reviewer's Responses to Questions

**Comments to the Authors:**

Reviewer #1: I appreciate the authors' efforts to address my and the other reviewers' questions, and I have no further comments.

Reviewer #3: The comments of all referees have been properly addressed, I thus recommend acceptance.

**Have the authors made all data and (if applicable) computational code underlying the findings in their manuscript fully available?**

Reviewer #1: Yes

Reviewer #3: Yes

PLOS authors have the option to publish the peer review history of their article (what does this mean?). If published, this will include your full peer review and any attached files.

Reviewer #1: No

Reviewer #3: No

---

## [Editor Report · Acceptance letter]

20 Nov 2023

PCOMPBIOL-D-23-00089R2 

GENERALIST: A latent space based generative model for protein sequence families

Dear Dr Dixit,

I am pleased to inform you that your manuscript has been formally accepted for publication in PLOS Computational Biology. Your manuscript is now with our production department and you will be notified of the publication date in due course.

With kind regards,

Anita Estes
